# Evolution recovers the fitness of *Acinetobacter baylyi* strains with large deletions through mutations in deletion-specific targets and global post-transcriptional regulators

Isaac Gifford [ID][‡], Gabriel A. Suárez[‡], Jeffrey E. Barrick [ID]*

Department of Molecular Biosciences, Center for Systems and Synthetic Biology, The University of Texas at Austin, Austin, Texas, United States of America

‡ These authors contributed equally to this work.
* jbarrick@cm.utexas.edu

**Data Availability Statement:** All relevant data are within the manuscript and its Supporting

## Abstract

Organelles and endosymbionts have naturally evolved dramatically reduced genome sizes compared to their free-living ancestors. Synthetic biologists have purposefully engineered streamlined microbial genomes to create more efficient cellular chassis and define the minimal components of cellular life. During natural or engineered genome streamlining, deletion of many non-essential genes in combination often reduces bacterial fitness for idiosyncratic or unknown reasons. We investigated how and to what extent laboratory evolution could overcome these defects in six variants of the transposon-free *Acinetobacter baylyi* strain ADP1-ISx that each had a deletion of a different 22- to 42-kilobase region and two strains with larger deletions of 70 and 293 kilobases. We evolved replicate populations of ADP1-ISx and each deletion strain for ~300 generations in a chemically defined minimal medium or a complex medium and sequenced the genomes of endpoint clonal isolates. Fitness increased in all cases that were examined except for two ancestors that each failed to improve in one of the two environments. Mutations affecting nine protein-coding genes and two small RNAs were significantly associated with one of the two environments or with certain deletion ancestors. The global post-transcriptional regulators *rnd* (ribonuclease D), *csrA* (RNA-binding carbon storage regulator), and *hfq* (RNA-binding protein and chaperone) were frequently mutated across all strains, though the incidence and effects of these mutations on gene function and bacterial fitness varied with the ancestral deletion and evolution environment. Mutations in this regulatory network likely compensate for how an earlier deletion of a transposon in the ADP1-ISx ancestor of all the deletion strains restored *csrA* function. More generally, our results demonstrate that fitness lost during genome streamlining can usually be regained rapidly through laboratory evolution and that recovery tends to occur through a combination of deletion-specific compensation and global regulatory adjustments.

Information files or publicly available from the NCBI Sequence Read Archive (PRJNA989175).

**Funding:** This work was supported by Welch Foundation grant F-1979 to J.E.B, National Science Foundation grant CBET-1554179 to J.E.B., National Science Foundation grant MCB-2123996 to J.E.B., a subcontract from the NSF BEACON Center for the Study of Evolution grant DBI-0939454 to J.E.B., and a UT Austin College of Natural Sciences Spark grant to J.E.B. The funders did not play any role in the study design, data collection and analysis, decision to publish, or preparation of the manuscript.

**Competing interests:** The authors have declared that no competing interests exist.

## Author summary

Genome streamlining reduces the complexity of organisms by eliminating large, non-essential portions of their genomes. This process occurs naturally in endosymbiont lineages and can be engineered to create microbial chassis that operate more efficiently and predictably. However, genome reduction often compromises the fitness of an organism when genes and combinations of genes are deleted that, while not essential, are advantageous. In this study, we used laboratory evolution to improve the fitness of a collection of *Acinetobacter baylyi* strains with large engineered deletions. In most cases, we found that spontaneous mutations could recover fitness lost due to deletions spanning many genes in these strains. These beneficial mutations were sometimes general, occurring in multiple strains and environments regardless of what genes were deleted, or specific, observed solely or more often in one environment or in strains with certain deletions. Mutations affecting proteins and small RNAs involved in post-transcriptional regulation of gene expression were especially common. Thus, recovering fitness often involves a combination of mutations that adjust global regulatory networks and compensate for lost gene functions. More broadly, our findings validate using laboratory evolution as a strategy for improving the fitness of reduced-genome strains created for biotechnology applications.

## Introduction

Genome streamlining is a process of genome reduction in which unnecessary or detrimental genes are lost or removed, resulting in a simpler genome. Gene loss occurs naturally in many microorganisms, including symbionts [1] and pathogens [2], that live in stable, nutrient-rich environments where genes encoding ancillary metabolic pathways are unnecessary. However, little is known about the detailed dynamics of natural streamlining, as most of our understanding of genome reduction comes from comparing the very different genomes of endosymbionts and their free-living relatives [3]. Two types of approaches have been used to experimentally study genome streamlining [4–7]. Top-down approaches are similar to natural processes in that they start with a free-living organism and sequentially delete non-essential pieces of its genome. This procedure has achieved various degrees of genome reduction in many bacterial and fungal species, including deleting up to 38.9% of the *Escherichia coli* genome [8]. The other approach, bottom-up design, assembles the DNA for a minimal chromosome *de novo* and then inserts it into a recipient cell to produce a synthetic organism. This procedure has been used to create *Mycoplasma mycoides* strains with extremely reduced genomes [9] and *Saccharomyces cerevisiae* strains with up to six and a half synthetic chromosomes [10].

It has been argued that streamlined genomes provide two major benefits to synthetic biology: removing genes that are detrimental in an environment of interest and reducing the complexity of cellular networks [4–7]. Most genomes encode pathways that are unnecessary in laboratory, bioreactor, or host environments; and these additional genes can negatively impact a desired phenotype. For example, genomes have been engineered to remove genes that cause autolysis [11] or block DNA uptake [12] to improve cell survival and competence, respectively. Other deletions, such as removing transposons, can also lower mutation rates [12–14]. Streamlining can be especially beneficial for metabolic engineering, when extraneous pathways redirect resources away from a desired product [7], as has been shown by deleting genes to increase antibiotic production in *Streptomyces avermitilis* [15] and recombinant protein production in *Pseudomonas putida* [16] and *Lactococcus lactis* [17]. Removing genes eliminates their interactions with remaining genes, simplifying modeling of metabolic and regulatory

networks, especially when genes of unknown function are deleted. Reducing cellular complexity in this way should make engineering more predictable [7].

The potential benefits of genome streamlining, however, may come at a cost. Removing what have been called "quasi-essential" genes [9] can result in fitness defects due to disrupting metabolic networks [18], reducing stress tolerance [8,14], or perturbing global gene expression [18], among other possibilities. Deletion of multiple, large genomic regions can also remove genes that individually have little impact on a cell's function but when removed together produce synthetic lethality [9,19] or are more deleterious than expected due to "synthetic-sick" interactions [14]. Because of our incomplete understanding of the consequences of deleting genes in top-down streamlined strains, laboratory evolution has been used as a means to recover lost fitness [14,18]. This strategy often works for single-gene deletions: they acquire mutations that compensate for the lost function [20,21]. Less is known about whether and when strains with many genes deleted at once can recover wild-type fitness through laboratory evolution. Short evolution experiments were able to partially restore the fitness of hypermutator strains of *Salmonella typhimurium* after they spontaneously accumulated mutations that included deletions of tens to hundreds of kilobases [22]. Different outcomes have been reported, however, after evolving strains of *E. coli* [18] and *Bacillus subtilis* [14] with cumulative engineered deletions comprising ~20–30% of their genomes. The *E. coli* strain wholly recovered fitness lost in minimal medium, primarily through rewiring metabolic and regulatory networks [18]. The *B. subtilis* strain exhibited more limited fitness recovery [14]. In light of these results, questions still remain about the impact of large deletions on an organism's fitness and how and whether laboratory evolution can recover lost fitness after this type of genome reduction.

Previously, we engineered strains of the metabolically versatile and naturally competent bacterium *Acinetobacter baylyi* ADP1-ISx with deletions of large genome segments that each encoded ≥17 nonessential genes [23]. In this study, we evolved replicate populations of eight reduced-genome strains of *Acinetobacter baylyi* ADP1-ISx in complex and defined medium laboratory environments. Deletions in these strains were constructed without examining the functions or expression levels of the genes being removed, which resulted in substantial fitness defects in some cases. Nevertheless, evolution restored fitness, at least partially, in most instances. Whole-genome sequencing revealed examples of both deletion-specific and shared genetic targets that recurrently evolved mutations in different populations. The most widespread mutations affected genes in a post-transcriptional global regulatory network. They appear to recapitulate or phenocopy a transposon insertion in the *csrA* gene in *A. baylyi* ADP1 that was removed during construction of the transposon-free ADP1-ISx progenitor strain. However, even in these cases, the effects of mutations within this network on gene function varied with the deletion strain and environment in which they evolved. Overall, our results validate laboratory evolution as a strategy for improving the fitness of reduced-genome strains, and they illuminate a path forward for engineering a minimal-genome *A. baylyi* strain.

## Results

### Multiple-gene deletion strains exhibit fitness defects

Previously, we created *Acinetobacter baylyi* strain ADP1-ISx [13] by deleting all transposable elements from the genome of the laboratory strain ADP1 [24]. Then, we examined possibilities for additional genome streamlining by attempting to construct 55 multiple-gene deletion (MGD) strains from ADP1-ISx [23]. Each of the 18 MGD strains that were successfully constructed has a single contiguous stretch of nonessential genes removed. For this study, we selected six of these MGD strains (MGD4, MGD6, MGD9, MGD12, MGD15, and MGD17)

with deletions ranging in size from 21.8 to 41.7 kb that removed from 17 to 47 genes (**Fig 1A**). We also examined a retained genome region (RGR) strain with a partially successful deletion, leading to loss of only six genes (RGR7), as an additional wild-type-like control, and two multiple-segment deletion (MSD) strains (MSD1 and MSD2) that were created by combining or expanding deletions in the original MGD strains. Strain MSD1 had the most-reduced genome, with a total of 293.5 kb and 268 genes removed, including a ribosomal RNA operon. MSD2 has a deletion of 69.8 kb that expands upon one of the original MGD segments (MGD11). It removes 67 genes.

To examine how the large deletions affected bacterial fitness, we constructed a GFP-expressing variant of ADP1-ISx. Then, we performed co-culture competition assays comparing this reference strain to each MGD strain and the ADP1-ISx progenitor in a complex medium (LB) (**Fig 1B**) and a defined minimal succinate medium (MS) (**Fig 1C**). The six MGD ancestor strains had fitness defects of 17.0% on average and up to 49.3% relative to ADP1-ISx in LB. The fitness defects were 10.0% on average and as high as 39.9% in MS. Four MGD ancestors (MGD4, MGD6, MGD9, and MGD12) had significantly lower fitness than ADP1-ISx in LB, and three of these four (MGD4, MGD6, and MGD9) also had significantly lower fitness than ADP1-ISx in MS (Benjamini-Hochberg adjusted $p < 0.05$, Welch's $t$-tests, see **Methods**). Two MGD ancestors (MGD4 and MGD6) exhibited significantly different fitness defects in the two different culture media (adjusted $p < 0.05$). For the MGD4 ancestor, the fitness defect was larger in MS by 18.3%, while the fitness defect of the MGD6 ancestor was 32.0% greater in LB.

## Multiple-gene deletion strains recover fitness during laboratory evolution

We evolved replicate populations of each of the six MGD, one RGR, and two MSD strains plus ADP1-ISx controls for 30 days (~300-generation) by performing 1000-fold serial-dilution transfers in either LB or MS. We began the experiment with six replicates of each ancestor strain containing a deletion in each environment with additional replicates of the ADP1-ISx controls. At the conclusion of the evolution experiment, we plated each population and picked a single large colony for further characterization.

To understand how fitness changed during the evolution experiment, we compared the results of co-culture competition assays pitting each of the six endpoint MGD clonal isolates against the GFP-expressing ADP1-ISx reference strain (**Fig 2**). The ADP1-ISx progenitor already had relatively high fitness in LB. Its six LB-evolved endpoint clones that we tested exhibited very little change in fitness in this environment (+0.1% mean ± 3.3% standard deviation), and the change due to evolution was not significant for any one clone (adjusted $p > 0.05$). The six MS-evolved ADP1-ISx clones whose fitness we assayed gained fitness overall (+13.4 ± 13.6%), but the difference in any one clone was, again, not statistically significant (adjusted $p > 0.05$). The evolved MGD strains exhibited wider variation in their fitness changes. Overall, they exhibited fitness increases of 7.9% on average and up to 25.1% in LB and 18.5% on average and up to 38.0% in MS. At least three of the six evolved clones derived from the same MGD ancestor had significant fitness increases (adjusted $p > 0.05$) for three MGD strains in LB (MGD4, MGD6, and MGD12) and for two MGD strains in MS (MGD6 and MGD9).

We expected that, in general, MGD strains that had the lowest initial fitness would gain the most fitness during the evolution experiment. Often this was the case, but there were also exceptions (**Fig 3**). In LB, most MGD strains followed the expected trend, but MGD9 was an outlier. Despite initially having the third-largest fitness defect in LB relative to ADP1-ISx from its deletion, there was a negligible change in the fitness of its six LB-evolved descendant strains

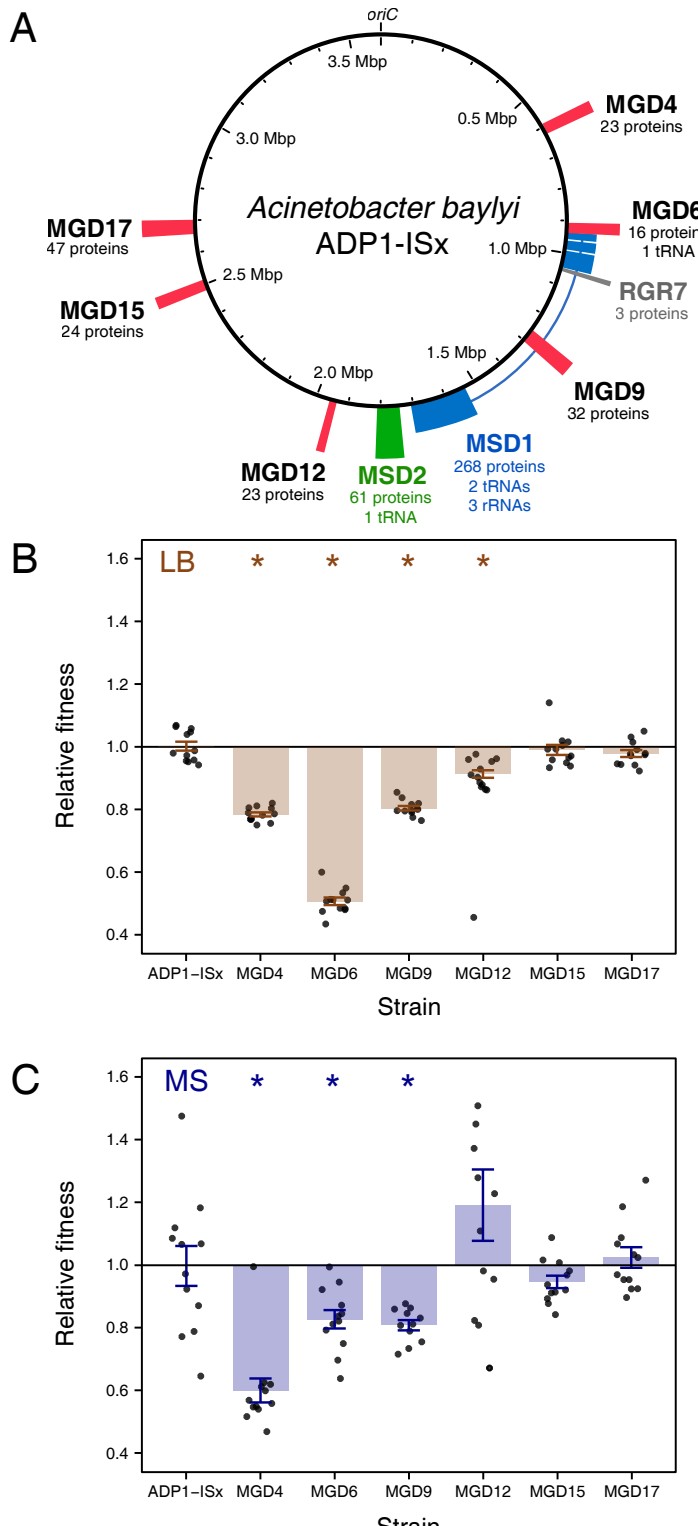

**Fig 1. Some reduced-genome variants of *A. baylyi* ADP1-ISx have fitness defects.** (**A**) *A. baylyi* ADP1-ISx chromosome showing the locations and sizes of regions removed in six multiple-gene deletion (MGD) strains, one retained-genomic region (RGR) strain, and two multiple-segment deletion (MSD) strains used as ancestors for the evolution experiment. The numbers of protein-coding, tRNA, and rRNA genes deleted in each strain are indicated. (**B, C**) Fitness of ADP1-ISx and the six MGD strains relative to a GFP-expressing variant of ADP1-ISx. Filled bars are

means. Error bars are 95% confidence intervals. Starred values are significantly different from ADP-ISx (Benjamini-Hochberg adjusted Welch's *t*-tests).

(−1.1 ± 4.7%). In MS, strains that evolved from MGD4, the ancestor with the greatest initial fitness defect of 39.9% in this medium, did not exhibit any increase in fitness (−2.0 ± 8.9%). MGD12 was an outlier in the opposite direction in MS. The ancestor strain had a 19.2% higher fitness than ADP1-ISx in this medium due to mutations that occurred during its construction (see below), yet the further increase in fitness of its six MS-evolved endpoint clones was also among the highest of all MGD strains (+33.5 ± 14.2%). Overall, these results show evolution was successful at improving the fitness of most MGD strains in most environments, though there were notable exceptions for two strains, each in a specific environment.

## Rates of genome evolution were similar for all deletions and in both environments

We sequenced the genomes of the MGD, RGR, and MSD ancestor strains and endpoint clones from independent replicate populations to understand how they evolved. Some evolved clones shared mutations that arose during strain construction or evolved in the cultures from which

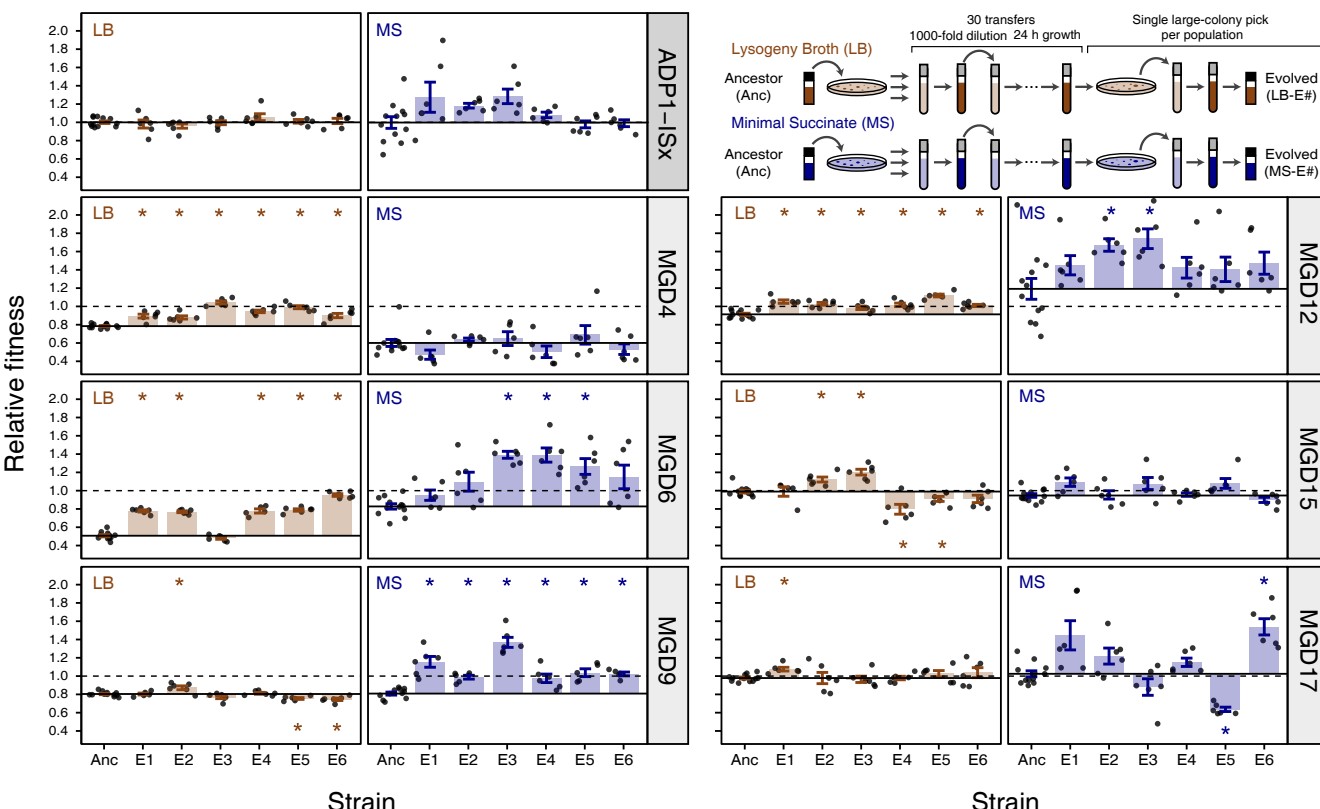

**Fig 2. Evolution of fitness in the evolution experiment.** The inset in the upper right shows the design of the evolution experiment and how final endpoint clonal isolates from each replicate population were selected for characterization. The other panels show the fitness determined for each of six evolved endpoint isolates (numbered 1–6) and their respective ancestor (Anc) relative to a GFP-expressing variant of ADP1-ISx. Points are the results of individual replicates of co-culture fitness assays. Horizontal solid lines are the average fitness of the respective ancestral strain. Horizontal dashed lines at a value of one are the relative fitness of the ADP1-ISx progenitor because this reference comparison was used to normalize all relative fitness measurements. Filled bars are means. Error bars are 95% confidence intervals. Starred values are significantly different from ADP1-ISx (Benjamini-Hochberg adjusted Welch's *t*-tests).

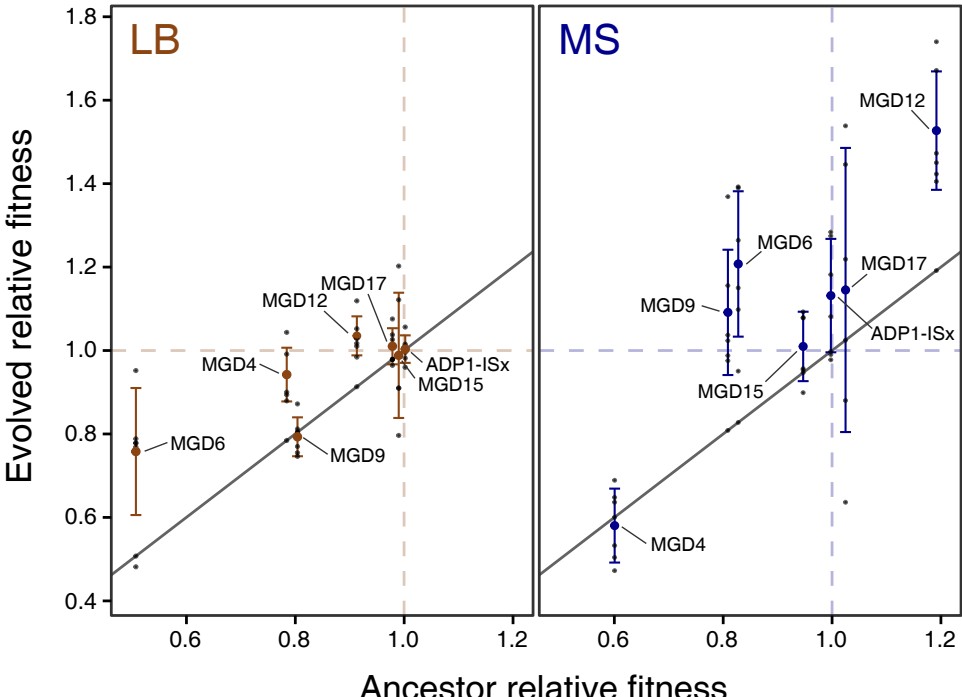

**Fig 3. Fitness of evolved clones versus their ancestors.** The relative fitness of each evolved endpoint clone that was characterized is plotted against the relative fitness of its ancestor. Fitness values measured for independently evolved clones from different replicate populations of the evolution experiment (small gray points) are summarized as means (larger colored points) with error bars for the standard deviations of the clone values. Points above the reference line with a slope of one represent improvements during evolution.

colonies were picked to begin each replicate population (**Fig 4**). Multiple ancestral strains and/ or cultures had pre-existing mutations in the same genomic region upstream of *ACIAD2521*, which encodes a putative divalent metal transporter. The MGD4 and MGD15 strains each had different mutations in this region, and the exact same mutations appear to have also evolved in subpopulations of cells in the initial MGD17 culture and in one culture of ADP1-ISx that was used to initiate six specific replicate populations (numbered 7–12). These and other mutations nearby in the same intergenic region were observed in other endpoint clones evolved in LB, suggesting that they both arise at high rates and are beneficial in this environment.

All other pre-existing mutations were restricted to one ancestor strain and its derivatives. The MGD12 ancestor had two such mutations, one in *murB* and one in *rpoD*. RpoD is the major housekeeping sigma factor ($\sigma^{70}$), and this mutation was an in-frame deletion of one CAG unit from a $(CAG)_3$ repeat that is likely a mutational hotspot. Interestingly, this mutation appears to have reverted in all the MGD12 clones that evolved in LB, an outcome that is consistent with how this deletion ancestor had lower fitness in LB but unexpectedly exhibited 19.2% higher fitness in MS (**Fig 1C**), though this difference in fitness did not reach statistical significance in our measurements (adjusted $p = 0.23$). The engineered deletion in MGD12 overlapped the promoter region of *murB*, which encodes an enzyme involved in peptidoglycan biosynthesis that is essential in *A. baylyi* ADP1 [23,25]. The MGD12 ancestor and all strains that evolved from it shared a synonymous substitution in the third codon of this gene that may restore its expression. The same nonsynonymous mutation in *ACIAD1813*, a putative transporter that may be related to phenol metabolism, was present in the MSD2 ancestor and all its descendants. RGR7 and its evolved clones shared a base substitution in the intergenic region

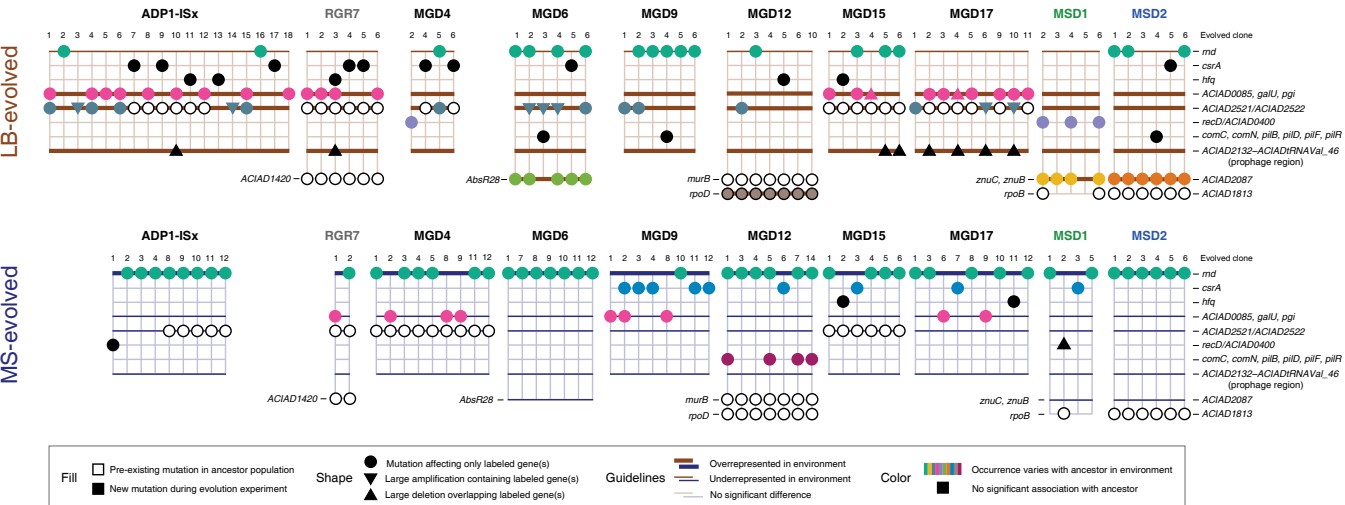

**Fig 4. Mutated genes associated with certain ancestors and environments.** Each symbol on the grids represents a mutation affecting the labeled gene, intergenic region (if two gene names are separated by a slash), any one of a set of labeled genes (if gene names are separated by commas), or a multigene region (if gene names are separated by a dash). Triangles are used for larger mutations that include the corresponding genes as well as their surrounding regions. Unfilled symbols are used for pre-existing mutations that fixed in the ancestor strain or were present in cultures used to start the evolution experiment. They were not included in the association analysis. Only genes or gene groups affected by mutations in three or more evolved isolates are shown. The bottommost lines of the grid show genes that were mutated only in the context of that one specific ancestor strain. A full list of evolved mutations is included in **S4 Table**. Genes or gene sets in which the rate of mutations differed depending on the culture environment are indicated by highlighting horizontal guidelines using thick (enriched) and thin (depleted) lines colored by the medium condition. Mutations with rates that differed depending on the ancestor strain within each environment are highlighted in color, with a different color used for each gene or gene set. In both cases the statistical significance of these associations was tested by using binomial regression models (Benjamini-Hochberg adjusted $p < 0.05$, likelihood ratio tests, see **Methods**).

upstream of *ACIAD1420*, a putative carbonate dehydratase. Finally, the same intergenic base substitution upstream of *rpoB*, which encodes the β subunit of RNA polymerase, was observed in one of the two sequenced MSD1 ancestors and two of the nine clones that evolved from it, suggesting that it was present in a subpopulation of cells in the founding culture.

Not counting these pre-existing mutations, at least one mutation and as many as nine mutations occurred in the lineages leading to the 140 sequenced endpoint clones from the evolution experiment. There were 1.86 mutations, on average, in the 73 LB-evolved clones and 1.36 mutations in the 67 MS-evolved clones. The difference in the rate at which mutations accumulated in the two environments was significant ($p = 0.022$, likelihood-ratio test comparing Poisson regression models), whereas there was not a significant effect of the ancestor strain on the rate ($p = 0.99$) or an ancestor by environment effect ($p = 0.70$). This held true even when comparing the very similar ADP1-ISx and RGR7 ancestors as one group against all MGD and MSD ancestors together ($p = 0.92$ for ancestor group effect and $p = 0.96$ for ancestor group by environment effect). In summary there were typically one or two mutations in each endpoint clone from the evolution experiment, with more on average in the LB-evolved clones.

## Mutations in specific genes are associated with certain environments and deletions

We next examined whether mutations affecting certain genes were associated with specific ancestral deletions or culture environments (**Fig 4**). We first tested for environment specificity by determining whether we could reject the hypothesis that the occurrence of mutations in a gene or set of genes was the same for endpoint clones that evolved in LB and MS. Then, within each environment, we determined whether we could reject the hypothesis that the rates of mutations in a gene or gene set were equal in clones that evolved from different ancestors. In

each case we determined whether the environment or ancestor effects were significant by comparing binomial regression models fit to the occurrence of mutations in each gene or gene set (Benjamini-Hochberg adjusted $p < 0.05$, likelihood ratio tests, see **Methods**).

We found that mutations in seven genes or sets of related genes were more likely in LB than in MS (**Fig 4**). One of these was the region upstream of the *ACIAD2521* divalent metal transporter gene that had pre-existing mutations in many ancestral strains (see previous section), even though these instances were not included in the association analysis. Mutations in this target were also unevenly distributed across ancestor strains. Similarly, mutations in *ACIAD0085*, *galU*, and *pgi*—three genes grouped together because they are part of the same extracellular polysaccharide (EPS) biosynthesis operon—and a deletion of a 49-kb prophage region that extends from *ACIAD2132* to *ACIADtRNAVal_46* were both significantly more common in LB. Mutations in the three EPS biosynthesis genes were significantly more likely to be mutated in clones that evolved from certain ancestors in both environments. In LB, these mutations were concentrated in strains that evolved from ADP1-ISx, RGR7, MGD15 and MGD17. In MS, most were in MGD4 and MGD9 and MGD17. The prophage deletions only occurred in LB and were mostly found in clones that evolved from the MGD15 and MGD17 ancestors, though the overall ancestor-specific association was only marginally significant after adjusting for multiple testing in this case (adjusted $p = 0.057$).

The four remaining targets that were significantly more likely to be mutated in LB-evolved clones were also significantly, and solely, associated with specific ancestors in this environment (**Fig 4**). In all seven MGD12 clones evolved in LB, mutations in *rpoD* reverted the in-frame deletion in a CAG trinucleotide repeat present in this ancestor strain (see previous section). All six LB-evolved MSD2 clones had nonsynonymous base substitutions in the pyridoxal phosphate-dependent aminotransferase *ACIAD2087*. Five of six LB-evolved MGD6 strains had base substitutions within or near a predicted homolog of the *AbsR28* small RNA of unknown function first experimentally detected in *Acinetobacter baumanii* [26]. Four of six MSD1 endpoint clones evolved in LB had mutations in genes encoding two components of a zinc ABC transporter found in the same operon (*znuB* and *znuC*). One of these mutations is a single-base deletion that results in a frameshift in *znuC*, which suggests that loss of this transporter's function is beneficial in the MSD1 deletion background.

Ribonuclease D (*rnd*) was the most commonly mutated gene in the entire evolution experiment. It was also the only gene that was significantly more likely to be mutated in MS than in LB (**Fig 4**). In MS, 46 of 67 endpoint clones (68.7%) had an *rnd* mutation, including at least one clone derived from every ancestor. Mutations in *rnd* were also common in LB, just less so: 19 of 73 clones (26.0%) evolved in this environment had an *rnd* mutation. These widespread mutations in *rnd* were also significantly associated with certain ancestors in both environments. Many *rnd* mutations were nonsense mutations or small insertions or deletions (indels) that are expected to completely inactivate the function of this gene (see next section).

While not significantly associated with either culture medium, mutations in the carbon storage regulatory protein *csrA* were associated with certain ancestors in MS, particularly MGD9 descendants that evolved in this environment. Mutations in *csrA* were either single-base substitutions or in-frame indels. They probably alter the function of this gene versus resulting in complete loss of function. A transposon that truncates the CsrA protein in *A. baylyi* relative to other *Acinetobacter* species was deleted during the creation of ADP1-ISx [16]. (**S2 Data File**). It is possible that the *csrA* mutations we observe have similar effects on its function to this transposon insertion. Mutations in glucose-6-phosphate isomerase (*pgi*) that include frameshifting indels were significantly associated with certain ancestors in LB, mostly ADP1-ISx and RGR7. LB-evolved mutations between the convergently oriented *recD* and *ACIAD0400* genes were also significantly biased between ancestors, with three of five MSD1

isolates containing mutations in this region, which was also mutated in one ADP1-ISx-MS clone and one MGD4-LB clone. Four of these mutations are within and the two others would completely delete a computationally predicted small RNA of unknown function [27]. Mutations in genes involved in natural competence (*comC*, *comN*, *pilB*, *pilD*, *pilF*, and *pilR*) were significantly associated with the MGD12 ancestor in MS. All seven of the mutations observed in these genes are insertions or deletion and all but one of them introduces a frameshift in the affected gene.

## Loss-of-function mutations in RNAse D were widespread

There were 65 total mutations in RNase D (*rnd*) across all evolved clones, which allowed us to examine their effects on gene function with more granularity. These mutations include nonsynonymous mutations, which may preserve some level of *rnd* function, and nonsense mutations, small indels, and large deletions, most of which are likely to result in complete loss of *rnd* function (**Fig 5A**). As noted above, mutations in *rnd* were significantly more common in MS-evolved strains than LB-evolved strains. Mutations in *rnd* in each environment were also heavily biased in different ways: the odds that an *rnd* mutation in MS was of a type that is likely to result in complete loss of function versus a nonsynonymous change was 8.0 times that in LB ($p = 0.00068$, Fisher's exact test). Some individual strains showed noticeable deviations from this general trend. For example, MGD9 was the only ancestor to have fewer *rnd* mutations in MS-evolved clones, and the spectrum was also reversed relative to other ancestors such that most of the *rnd* mutations observed in its LB-evolved clones were putative loss of function mutations.

## Effects of deleting *rnd* on fitness vary with ancestor and environment

To directly study the impact of RNAse D knockout on the different MGD strains, we deleted the *rnd* gene in the unevolved ancestor of each MGD strain and ADP1-ISx. Then, we used co-culture competition assays to measure the fitness effects of *rnd* deletion in both LB and MS environments (**Fig 6**). Overall, across all strain backgrounds, deletion of *rnd* increased fitness by 7.4%, on average, in MS-evolved strains, and this difference was significant ($F_{1,76} = 0.37$, $p = 0.016$). In LB, it increased fitness by 1.2%, on average, which was not significant ($F_{1,76} = 6.1$, $p = 0.54$). The higher fitness benefit in MS is in accordance with the observation of more mutations, including more loss-of-function mutations, in clones that evolved in MS (**Fig 5**).

Most strains did not individually show a significant change in fitness in either medium after deleting *rnd* (Benjamini-Hochberg adjusted $p > 0.05$, *t*-tests) (**Fig 6**). The exceptions were MGD12, in which fitness decreased by 19.6% in LB (adjusted $p = 0.0011$); and MGD6, in which fitness increased by 33.8% in MS (adjusted $p = 0.0070$). Mutations in *rnd* were rare in MGD12 strains evolved in LB (**Fig 5**), which makes sense given the deleterious effects of deleting *rnd* in MGD12 in this environment. Similarly, there were no *rnd* mutations in LB-evolved MGD17 strains, which is in agreement with the apparent, though not statistically significant, reduction in fitness upon deleting *rnd* in MGD17. Overall, these results suggest that deleting RNase D is beneficial in many but not all combinations of ancestors and environments, in agreement with how putative knockout mutations are widespread, particularly in MS-evolved clones, but other types of mutations commonly evolved in certain strains.

## Convergent overlapping duplications evolved in some strains

Large duplications evolved in five strains: three MGD6, one MGD1, and one ADP1-ISx strain. The duplicated regions include the *ACIAD2521/ACIAD2522* intergenic region that had pre-existing mutations in many ancestor strains and commonly experienced new mutations

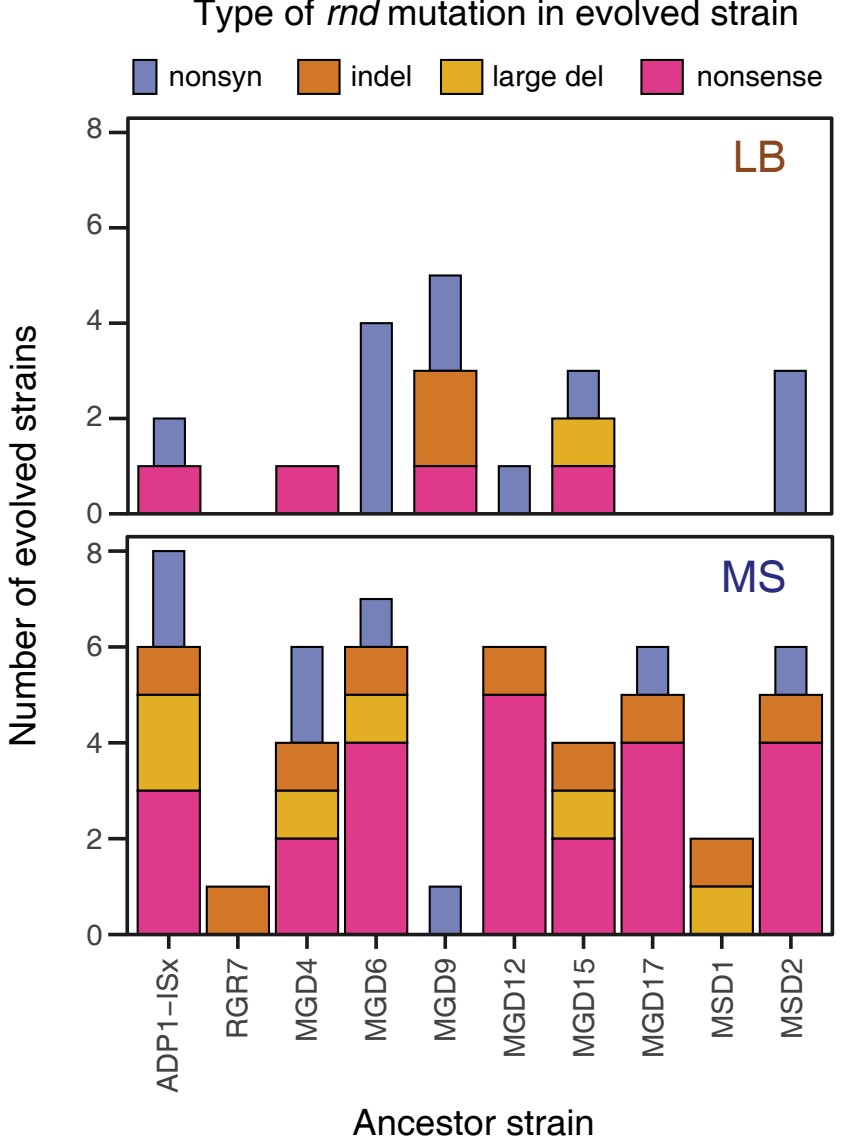

**Fig 5. Spectra of RNase D mutations vary with ancestor and environment.** Wider bars are used for types of mutations that are likely to result in complete loss of gene function.

during the evolution experiment in strains that evolved in the LB environment (**Fig 4**). Four of these five large duplications were also in LB-evolved strains. All four had significant overlap and included the genes from *ACIAD2501* to *ACIAD2531*. This shared region is enriched in genes involved in glutamate and glutamine synthesis (false discovery rate, FDR = $2.2 \times 10^{-6}$), ketone body metabolism (FDR = $2.2 \times 10^{-6}$), pyrroloquinoline quinone biosynthesis (FDR = $1.3 \times 10^{-5}$), and valine/leucine/isoleucine degradation (FDR = 0.012).

## Discussion

We examined fitness recovery of *Acinetobacter baylyi* ADP1-ISx strains engineered to have reduced genomes during ~300-generation evolution experiments in complex and defined growth media. Several of the engineered single deletions significantly decreased fitness in one

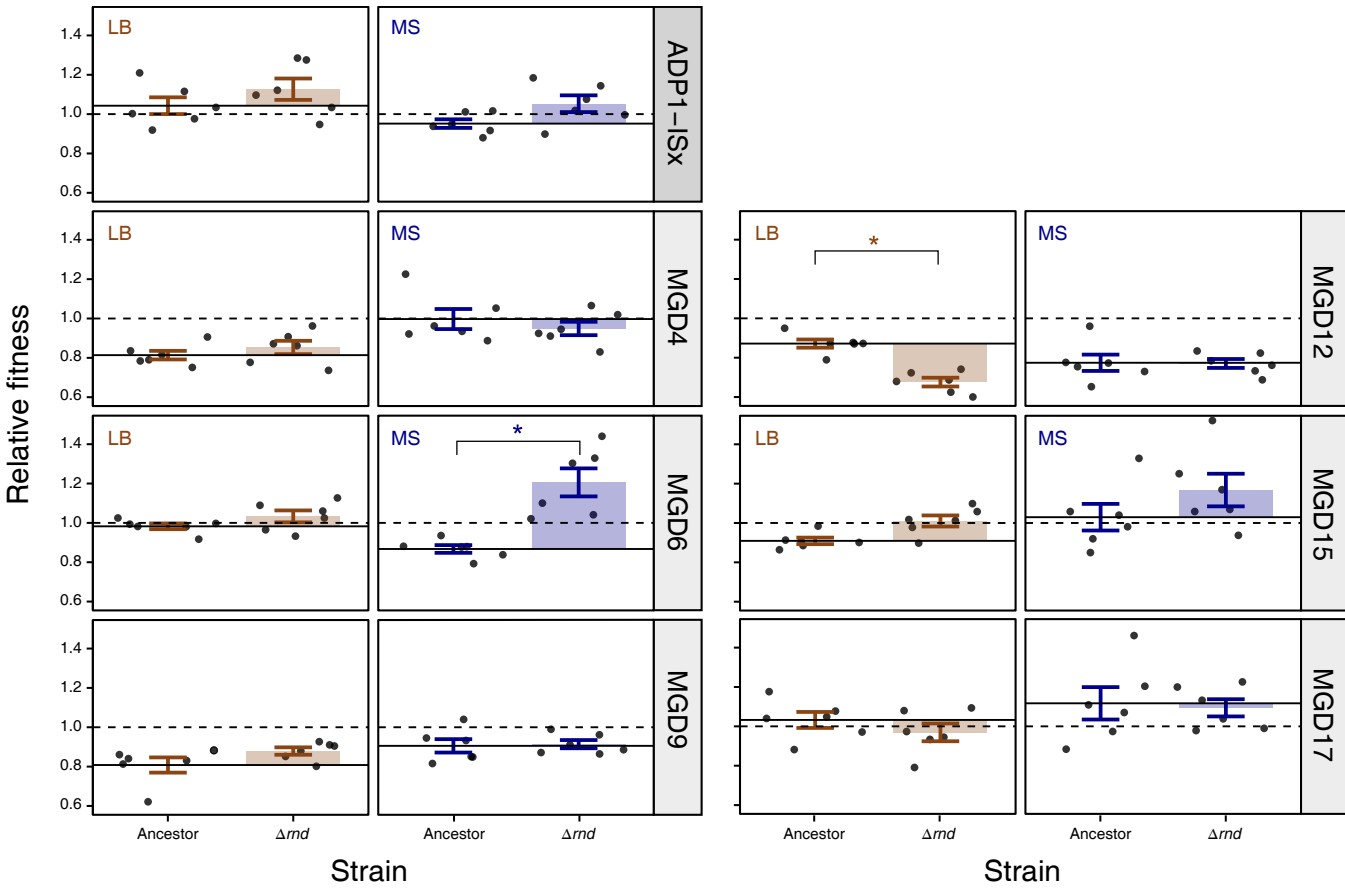

**Fig 6. Effects of deleting RNase D in different ancestors on fitness in each environment.** Each panel shows the fitness of an RNAse D deletion strain (Δ*rnd*) and its ancestor (Ancestor) relative to GFP-expressing ADP1-ISx. Points are the results of individual replicates of co-culture fitness assays. Horizontal solid lines are the average fitness of the respective ancestral strain measured in these competitions. Horizontal dashed lines at a value of one are the relative fitness of the ADP1-ISx progenitor determined in a prior set of experiments, which was used as a reference value for normalizing all measurements in these assays. Error bars are 95% confidence intervals. Starred comparisons indicate a significant difference between the fitness values measured for an ancestor and its Δ*rnd* mutant (Benjamini-Hochberg adjusted $p < 0.05$, *t*-tests).

or both media. Following evolution, most of the resulting strains had gained fitness, but two failed to show significant recovery in one of the two environments (MGD9 in LB and MGD4 in MS). In these cases, the engineered deletion seems to have eliminated one more genes with a function that cannot be fulfilled by mutations altering the expression and activities of the remaining genes, at least through evolutionary pathways that were accessible in this short experiment.

In the more common cases where adaptation was able to recover lost fitness and even surpass the fitness of the ancestral strains, certain genes were recurrently mutated in specific environments and in specific deletion strain backgrounds, likely because these mutations compensate for lost functions. In the defined medium (MS), mutations in RNase D (*rnd*) were overwhelmingly common, whereas mutations in a variety of pathways were overrepresented in the complex medium (LB). The strongest associations were between the *AbsR28* small RNA and MGD6 in LB, *rpoD* and MGD12 in LB (and a pre-existing *rpoD* mutation in MS), *znuC* or *znuB* and MSD1 in LB, and *ACIAD2087* and MSD2 in LB. Other significant associations included enrichment of mutations in a putative small RNA located in the *recD/ACIAD0400*

intergenic region in MSD1 in LB and various genes involved in natural competence in MGD12 in MS.

*A. baylyi* ADP1 has been evolved in the laboratory in several prior experiments [28–30]. Mutations in genes involved in competence (*comC*, *comN*, *pilB*, *pilD*, *pilF*, *pilR*) and extracellular polysaccharide biosynthesis (*ACIAD0085*, *galU*, and *pgi*) are similar or identical to those observed previously [29]. Their association with certain ancestors and environments in our study may be less because they are only beneficial in certain circumstances and more that they are generally beneficial in *A. baylyi* ADP1 but are outcompeted by other, more-beneficial mutations in many combinations of deletion ancestors and environments [31,32].

Two of the most commonly mutated genes (*csrA*, and *hfq*) are global post-transcriptional regulators of gene expression that interact with small RNAs [33,34]. The canonical function of RNase D (*rnd*), the gene that experienced the most mutations, is to process tRNAs [35]. These mutations could have widespread effects on gene expression by altering the availability of mature tRNAs for translation. RNase D has also been shown to degrade small RNAs [36], which means that *rnd* mutations could also more directly impact the same processes as *csrA*, and *hfq*. Mutations in *rnd* were more common in MS, and mutations in *csrA* were significantly associated with certain ancestors, like MGD9, in MS. There is also clearly a genetic background effect that causes different types of mutations (point mutations versus knockout mutations) in *rnd* to be more or less beneficial to fitness in the context of different ancestral deletions. This conclusion is supported both by statistical trends in mutation types and direct measurements of the fitness effects of deleting *rnd* in different strain backgrounds. Yet, mutations in both *rnd* and *csrA* evolved in at least one descendant of every engineered deletion stain and their ADP1-ISx progenitor, so—despite the idiosyncratic effects of specific mutations—altering this regulatory network in some way is widely beneficial.

During creation of the transposon-free ADP1-ISx strain [16], we restored the full-length sequence of *csrA* that was interrupted near its C-terminus by an IS*1236* insertion in *A. baylyi* ADP1. The widespread mutations that we observed in genes involved in post-transcriptional regulation in ADP1-ISx, including *csrA*, suggest that this earlier transposon insertion was beneficial and that we are observing multiple ways that evolution can reoccur to similarly alter global gene regulation in *A. baylyi*. Rnd, CsrA, and Hfq either interact with or can modulate the activity of small regulatory RNAs. Thus, the mutations that we observed in two small RNA genes of unknown function (*AbsR28* and *recD*/*ACIAC0400*) could also be part of the same regulatory network. Future work, reconstructing these mutations and measuring their effects on gene expression and fitness, alone and in combination, could validate our statistical correlations and map the genetic interactions in this network that are important for robust *A. baylyi* growth.

Recently, adaptive evolution of the streamlined *E. coli* strain MS56 revealed global transcriptional changes in gene expression due to mutations in the housekeeping sigma factor (*rpoD*) that alter its promoter specificity [18]. These types of global changes in transcription make sense in the context of the increased expression of genes controlled by the stationary phase sigma factor (*rpoS*) that are observed in other reduced-genome *E. coli* strains (MDS42 and MDS69) [37]. Similarly, mutations in the transcriptional machinery (e.g., *rpoC*) are often observed as a means to optimize metabolism and improve growth in many adaptive laboratory evolution experiments with standard *E. coli* strains [38–40]. We observed frequent mutations affecting post-transcriptional regulation. Thus, it appears that re-tuning global regulatory processes (e.g., transcription, translation) is often the most expeditious and generalizable way for evolution to improve the fitness of both wild-type genomes and less-fit reduced genomes. Prior evolutionary studies of reduced-genome *E. coli* observed changes in translation rather than post-transcriptional regulation [18], and there was little fitness recovery for reduced-

genome *B. subtilis* strains in a complex medium [14], suggesting that mutations in different global processes may be more beneficial in different organisms with different sets of genes deleted.

Future work constructing a minimal *A. baylyi* genome could progress by combining the deletions we describe into a single strain. Some deletions, such as those in MGD15 and MGD17, that did not deleteriously impact fitness should be easy to combine. Deletions with fitness costs, such as MGD6 and MGD12, could be added along with compensatory mutations that evolved in the corresponding ancestor strains in this study. The natural competence of *A. baylyi* can be used to streamline this process and test multiple compensatory mutation candidates at the same time. Several past studies have PCR amplified DNA fragments with beneficial alleles from evolved *A. baylyi* strains, mixed them, and provided this DNA back to evolving populations [13,28,41]. However, it is possible that these mutations may no longer be beneficial in the context of multiple-deletion strains, so this approach needs to be experimentally validated.

Strains with certain deletions (MGD4 and MGD9) had large fitness defects and were unable to recover fitness following evolution. This result makes removing either of these regions of the chromosome a dead end in terms of incorporating these deletions into a minimal genome *A. baylyi* strain that robustly functions in the laboratory. Minimal genome engineering projects in other organisms have avoided deleting genes that, while not essential, are required for robust growth, so-called "quasi-essential" genes [9], and purposefully left highly expressed genes untouched [18]. Because the ADP1-ISx deletions studied here did not differentiate the targeted genes by any criteria beyond essentiality when they were planned and constructed, the deletions that did not recover growth likely contained some of these types of genes. Further streamlining of *A. baylyi* genomes to include these deletions and others that could not be constructed [23] would require debugging, for example, by adding back any "irreplaceable" genes that are needed to restore viability and fitness in ways that evolution on laboratory timescales cannot achieve.

Despite encountering some limitations, the overwhelming result of our study is that evolution is useful for mechanistically probing why there are fitness defects after specific gene deletions and for recovering the fitness of reduced-genome strains. Thus, it should be considered a critical component of strain engineering campaigns that are creating the next generation of simplified, efficient, and more predictable chassis organisms for biotechnology applications. Given that microbes that naturally evolve extremely reduced genomes generally become less robust [1], it remains to be seen how far these endeavors that seek to build, remodel, and evolve streamlined genomes can be pushed, whether their goals are to optimize microbial cell factories or understand the essential genetic architectures of cellular life.

## Materials and Methods

### Culture conditions

*A. baylyi* cultures and plates were incubated at 30˚C. Liquid cultures were grown in the Miller formulation of lysogeny broth (LB) or minimal medium for *Acinetobacter* supplemented with 25 mM succinate (MS) [25]. Solid media were prepared by adding 1.5% (w/v) agar. Liquid cultures were grown in 18×150 mm test tubes with 200 r.p.m. orbital shaking over a 1-inch diameter in 5 ml of medium unless otherwise specified. Frozen stocks of strains and evolved populations were stored at –80˚C with 15–20% (v/v) glycerol. Where appropriate, media were supplemented with 50 μg/ml kanamycin (Kan) or 200 μg/ml 3′-azido-2′,3′-dideoxythymidine (AZT).

## Strain construction

MGD and RGR strains were created previously using Golden Transformation [23]. To construct the ancestor strains used in the evolution experiment, we removed the dual-selection *tdk-kanR* cassette from their genomes using "rescue" Golden Transformations that leave behind a 4-bp scar in place of the deleted genes [23]. The two MSD strains and mutant MGD strains with knockouts of the *rnd* gene were created for this study using the same two Golden Transformation steps, first to replace the deleted gene or region with the *tdk-kanR* cassette and then to remove the cassette. MSD1 was created using four of these editing cycles, one for each of its deletions, and it differed from the other constructions in using *tdk-kanR* integration and rescue constructs created using overlap extension PCR, which left no scars after genome editing. **S1 Table** describes the coordinates and gene content of the deletions present in each ancestor.

## Evolution experiment

To begin independent replicate populations, diluted cultures of each strain were first plated onto LB and MS agar to obtain single colonies. After growth, random colonies were picked into 1 mL liquid cultures in the same medium to grow initial T0 populations. On each of the following days of the evolution experiment, 5 μL of culture was transferred into 5 mL of fresh medium (a 1:1000 dilution) after 24 h of growth, except for some populations in MS medium that were transferred every 48 h at the beginning of the experiment until they began to reliably saturate in 24 h. This process was repeated for a total of 30 cycles of serial dilution and regrowth (~300 generations). Initial T0 and final T30 populations were archived as freezer stocks.

Dilutions of the final T30 populations were spread on LB or MS agar, matching the medium in which they had evolved. After overnight incubation, a single large colony was selected from each plate. The rationale for picking a large colony (versus a random colony) was that this choice might preferentially pick cells within a population that have evolved increased fitness. Each of the selected T30 endpoint clones was then grown overnight in 1 mL of the medium in which it was evolved and stored as a freezer stock before further characterization.

## Competitive fitness assays

We measured fitness by competing ancestor and evolved clonal isolates versus a GFP-tagged ADP1-ISx strain (ADP1-ISx-GFP) in co-culture, as described previously [29]. ADP1-ISx-GFP encodes a *gfp* gene derived from plasmid pBAV1K-T5-gfp [42] inserted in the intergenic region between *ACIAD2778* and *ACIAD2783*, in place of one of the IS elements deleted during the creation of ADP1-ISx. Briefly, 2 μl of the freezer stock of each strain was revived in LB or MS media overnight. The following day six replicate competition cultures per test strain were started by combining 1.5 μl ADP1-ISx-GFP culture and 3.5 μl of each test culture in 5 ml of fresh medium. These cultures were immediately diluted $10^3$-fold in sterile saline, and 50 μl of the dilutions were plated on LB agar. The competition cultures were incubated for 24 h and then diluted $10^6$-fold in sterile saline. From these dilutions, 50 μl was spread on LB agar. After overnight incubation at 30°C, we counted GFP$^+$ and GFP$^-$ colonies on the agar plates using a Dark Reader blue light transilluminator (Clare Chemical Research, Dolores, CO). Colony counts were used to calculate relative fitness as the ratio of Malthusian parameters [43,44]. To include competition assay replicates that had low or zero counts of one colony type, we added a pseudocount of 0.5 to the initial and final GFP$^+$ and GFP$^-$ colony counts in all competitions before calculating relative fitness.

Fitness assays were conducted in experimental blocks that included competing two different clonal isolates of each ancestral strain and all evolved strains derived from that ancestor versus ADP1-ISx-GFP, with six replicate competition cultures per pair of strains being tested. We performed all comparisons of relative fitness values using two-tailed Welch's *t*-tests and used the Benjamini-Hochberg procedure to correct *p*-values for multiple testing. ADP1-ISx was slightly more fit, on average, than the marked ADP1-ISx-GFP strain in both LB and MS, with a relative fitness of 1.053 and 1.049, respectively. This difference was significant in LB but not in MS ($p$ = 0.0044 and $p$ = 0.48, respectively). Because this fitness difference did not significantly vary between LB and MS ($p$ = 0.95), we normalized the relative fitness values we estimated in all competitions by dividing by 1.051, the mean relative fitness of ADP1-ISx to ADP1-ISx-GFP across all 24 competitions between those strains. This correction and all of our statistical analyses that compare strains that were not directly competed assume that our measurements of relative fitness are transitive. Colony counts, and raw and normalized relative fitness values for each replicate of these competition assays are provided in **S2 Table**.

To analyze the results of the evolution experiment, we first asked whether there was a difference in the relative fitness values measured for the two different isolates of each ancestor. We did not find a significant difference in fitness between the pairs of clonal isolates for any of the ancestors in either LB or MS media after correction for multiple testing across the 14 total comparisons (adjusted $p$ > 0.05). Therefore, all fitness measurements for each of the two ancestral clones were combined when making further comparisons. Next, we asked whether the relative fitness of each MGD ancestor was lower than that of ADP1-ISx in the same medium, correcting for multiple testing within this set of 12 comparisons. The same set of measurements was then used to test whether the fitness of each MGD ancestor strain relative to ADP1-ISx-GFP was different in LB versus MS, a set of 6 comparisons. Finally, we tested whether the relative fitness of each of the evolved strains increased in the medium in which it evolved compared to its ancestor. Here, we separately corrected *p*-values for multiple testing within each set of 6 comparisons between an ancestor and its evolved derivatives in a given medium.

To analyze the effects of *rnd* mutations on fitness in different ancestors and environments, we constructed variants of ADP1-ISx and each MGD ancestor strain with the *rnd* gene deleted, as described above. Then, we conducted competitions in experimental blocks that included each MGD ancestor strain and its *rnd* mutant. Again, all strains were competed against ADP1-ISx-GFP and the measured relative fitness values were corrected for the difference between the fitness of this reference strain and ADP1-ISx found in the prior set of competitions. Here, we corrected *p*-values for multiple testing across the entire set of 14 comparisons. Colony counts, and raw and normalized relative fitness values for each replicate of these *rnd* mutant competition assays are provided in **S3 Table**.

## Whole genome resequencing

Genomic DNA was isolated from 1–2 mL cultures of Day 0 and Day 30 clones grown from freezer stocks in the same medium in which they evolved using the PureLink Genomic DNA Mini Kit (Invitrogen) according to the manufacturer's instructions. DNA concentrations were measured using a Qubit Fluorometer (ThermoFisher). Dual-indexed Illumina sequencing libraries were constructed using 100 ng of genomic DNA as input into the 2S Turbo kit (Swift Biosciences) as per the manufacturer's instructions except that reaction volumes were conducted in 20% of the standard volume, the post-ligation bead clean up step was eluted directly into the PCR master mix, and the beads were left in the reaction during the 18 cycle PCR

reaction. Sequencing was performed at Macrogen (Rockville, MD) on an Illumina HiSeq X. FASTQ files for all 213 samples are available from the NCBI Sequence Read Archive (PRJNA989175).

## Mutation analysis

Adapter sequences were removed from reads using Trimmomatic (version 0.36) [45] in paired-end Illumina mode with options allowing 4 seed mismatches, a palindrome clip threshold of 30, and a simple clip threshold of 10. Mutations were called by using *breseq* (version 0.33.2) [46,47] to compare trimmed reads to a version of the *A. baylyi* ADP1 reference genome [24] corrected for mutations/discrepancies found in prior sequencing of the ADP1 isolate used by our lab [29]. We annotated the *AbsR28* small RNA gene [26] based on a match to Rfam [48] family RF02606 using Infernal [49]. As described previously [46], we used *gdtools* to simulate evolved genomes and re-ran *breseq* against these to check our mutation predictions. Runs with *breseq* in polymorphism mode showed that a few mutations were present in only a fraction of the sequenced cell populations, either because mixed colonies derived from multiple cells were selected when picking clones or due to evolution in cultures after picking clones. These mutations were assigned frequencies between 0 and 100% based on the relative numbers of reads supporting different alleles. We confirmed that the expected ADP1-ISx transposon deletions [13] and strain-specific deletions [23] were present in each ancestor and evolved genome. The annotated reference genome sequence and full details for all predicted mutations are provided in **S1 Data File**. Final lists of mutations, excluding the engineered deletions, in the evolved clones are provided in **S4 Table**.

After filtering out pre-existing mutations that were shared between ancestors and strains that evolved from them, we analyzed rates of genome evolution by fitting Poisson regression models to the counts of mutations in each endpoint clone. To account for polymorphic mutations, the frequencies of all predicted mutations in a clone were added together and then rounded down to the nearest integer. We used likelihood ratio tests comparing models that included different factors to examine whether the numbers of mutations in evolved clones significantly varied with environment, identity of the ancestor strain, or with environment-ancestor combinations.

We analyzed the counts of new mutations in different genes or sets of related genes for evidence that they were associated with different environments or ancestors using binomial regression models. This approach is similar to a Poisson regression framework proposed by others [50], but it assumes that a beneficial mutation will occur at most once in any given gene or related set of gene targets in each evolved clone. Only targets with at least three mutations across all evolved strains were analyzed. We performed three comparisons on each of the genes or sets of related genes in which we observed three or more mutations across all evolved clones. First, we tested whether a binomial model with environment-specific odds fit the occurrence data better than a model with uniform odds in both environments. Then, we tested whether binomial models with ancestor-specific odds fit significantly better than a model with uniform odds for all ancestors, separately within each of the two environments. For determining the significance of each type of association, we calculated *p*-values for likelihood ratio tests and corrected them multiple testing using the Benjamini-Hochberg procedure.

The CsrA sequence alignment in **S2 Data File** was created by downloading protein sequences from orthology group K03563 from the KEGG database [51] and aligning them with MUSCLE (version 3.8) [52]. We used STRING [53] to perform the functional enrichment analysis of genes found in the region of overlap shared by four large duplications in LB-evolved clones.

## Supporting information

**S1 Table. Deletion ancestor strain information.**
(CSV)

**S2 Table. Ancestor and evolved clone competition results.**
(CSV)

**S3 Table. Rnd deletion mutant competition results.**
(CSV)

**S4 Table. Evolved clone mutations.**
(CSV)

**S1 Data File. Reference genome and mutations in all sequenced strains.**
(ZIP)

**S2 Data File. Alignment of CsrA sequences from *Acinetobacter* genomes.**
(TXT)

## Acknowledgments

We thank Daniel Deatherage for preparing genome sequencing libraries, Sean Leonard for assisting with exploratory data visualization and analysis, and other members of the Barrick lab for helpful feedback. We acknowledge the Texas Advanced Computing Center (TACC) at The University of Texas at Austin for providing high performance computing resources.

## Author Contributions

**Conceptualization:** Isaac Gifford, Gabriel A. Suárez, Jeffrey E. Barrick.

**Data curation:** Isaac Gifford, Gabriel A. Suárez, Jeffrey E. Barrick.

**Formal analysis:** Isaac Gifford, Gabriel A. Suárez, Jeffrey E. Barrick.

**Funding acquisition:** Jeffrey E. Barrick.

**Investigation:** Isaac Gifford, Gabriel A. Suárez.

**Supervision:** Jeffrey E. Barrick.

**Visualization:** Isaac Gifford, Jeffrey E. Barrick.

**Writing – original draft:** Isaac Gifford, Gabriel A. Suárez, Jeffrey E. Barrick.

**Writing – review & editing:** Isaac Gifford, Gabriel A. Suárez, Jeffrey E. Barrick.

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
