## [Decision Letter · Decision Letter 0]

9 Aug 2024

Dear Dr Barrick,

Thank you very much for submitting your Research Article entitled 'Evolution recovers the fitness of *Acinetobacter baylyi* strains with large deletions through mutations in deletion-specific targets and global post-transcriptional regulators' to PLOS Genetics.

The manuscript was fully evaluated at the editorial level and by three independent peer reviewers. The reviewers appreciated the attention to an important topic but identified some minor concerns that we ask you address in a revised manuscript.

We therefore ask you to modify the manuscript according to the review recommendations. Your revisions should address the specific points made by each reviewer.

To resubmit, log into your Editorial Manager account and select the option 'Revise Submission' in the 'Submissions Needing Revision' folder.

Yours sincerely,

Xavier Didelot

Academic Editor

PLOS Genetics

Lotte Søgaard-Andersen

Section Editor

PLOS Genetics

Reviewer's Responses to Questions

**Comments to the Authors:**

Reviewer #1: This is an interesting and important paper. It reads very well. Genome reduction is an important process in bacterial function and evolution, particularly in the areas of endosymbiosis (where small genomes are an outcome of the process) and synthetic biology (where small genomes are the goal of the process). Understanding how cells adapt to large deletions, whether they be in nature or in the lab, is therefore important and of interest to a broad readership. I do not know of a paper that addresses this issue in an comprehensive manner as this paper. The finding that regulator non-functionalization is the primary means that cells use to compensate for large deletions is very cool.

I only have two small issues.

The first is a lack of citation of a classic related paper. On lines 104-105 in the introduction, the authors state that, “Less is known about whether and when strains with many genes deleted at once can recover wild-type fitness through laboratory evolution.” That’s true, there is not a ton known about this, but I think you are missing perhaps the most relevant example from the literature, Nilsson et al., 2005, Bacterial genome size reduction by experimental evolution. PNAS 102:12112-12116. Nilsson et al. used a mutator strain of Salmonella to make deletions (in a single-cell bottleneck regime) ranging in size from 1 to 202 kb, and then used experimental evolution (in large population sizes) to show that fitness could be quickly regained in many cases. It’s almost exactly what you see here, although Nilsson and colleagues were not able to sequence the genome, etc., leaving anything about genes and mechanisms of recovery unknown. But still, worth citing and discussing.

The second is related to lines 415-416, “Three of the most commonly mutated genes (rnd, csrA, and hfq) are all global post-transcriptional regulators of gene expression [32–34].” I think that is true for csrA and hfq, but I am not convinced that this statement is true for rnd. The function of RNase D is a bit elusive, and has been for a long time, but it is clear that it is a 3’ exoribonuclease. My understanding is that it is quite active on tRNA ends, and seems to be involved in 3’ tRNA maturation (although there are a suite of ribonucleases that have overlapping functions in tRNA processing). I agree that some papers, including citation 34 in this paper, suggest that it might be involved in processing other small RNAs in the cell (it most certainly does, most ribonucleases are not finely specific for one molecule). But processing a sRNA in Myxobacteria, a sRNA that is involved in development in that bacterium, does not mean that RNase D is in general a regulator of gene expression, in my view. I do think there is something quite interesting here in the data the authors are seeing related to rnd, but perhaps the strong signal you are seeing is related to tRNA processing or some other secondary global effect that looks like regulation (tRNA processing gets messed up, that changes what the cell is doing, you see a growth effect), but is not itself related to directly binding to transcripts to regulate transcription or translation.

Reviewer #2: The manuscript "Evolution recovers the fitness of Acinetobacter baylyi strains with large deletions through mutations in deletion-specific targets and global post-transcriptional regulators" is well-written and includes interesting findings. Here, are my general and specific comments/questions:

- Introduction and Discussion are very long, please condense.

- this might be beyond the scope of this work, but can you test if the gained fitness (evolved strains) is stable or reversible?

Lines 233-235: "The engineered deletion in MGD12 overlapped the promoter region of murB, which encodes an enzyme involved in peptidoglycan biosynthesis that is essential in A. baylyi ADP1". Does this deletion fuse the murB into another reading frame (that is to say as a result of frame-fusion it might use a secondary promoter)?

- Can you please more clearly explain difference in the evolution in MS versus LB and why? This is in the manuscript but I think it needs to be stated more clearly for readers.

- line 307: Did you check the reads to ensure this is not due to the sequencing/assembly errors?

- under "Whole genome resequencing" (Line 666): Please include your reads' QC steps, what programs/settings did you use?

Reviewer #3: The paper explores the potential for laboratory evolution to restore fitness in Acinetobacter baylyi ADP1-ISx strains with extensive genome deletions. By evolving these strains over roughly 300 generations in both minimal and complex media, researchers observed fitness improvements in most cases. Genome sequencing of the evolved strains revealed mutations associated with different ancestral strains and environments. This study demonstrates how laboratory evolution can restore fitness in strains with reduced genomes, providing valuable insights for synthetic biology applications aimed at developing streamlined microbial chassis.

I find the study design to be well-conceived and the paper clearly written within its chosen structure. The abstract, introduction, and discussion sections are particularly well-crafted and illuminating. The initial results on detected fitness restoration are also very clear.

However, I struggled with the central part of the paper, which discusses the sequencing results. There are pre-existing mutations, mutations that potentially co-occurred in the founding culture, and very few mutations that occurred during the actual evolution. Among these few de novo mutations that evolved in parallel, some are generally beneficial in the corresponding environment. It would be helpful if the paper focused more on the few background-specific mutations that occurred during the evolution experiment. These mutations seem most relevant to the core question of fitness restoration during evolution.

On a related note, considering the paper's focus on fitness restoration after different types of deletions, it seems slightly odd that the strongest adaptive signal highlighted in the abstract pertains to mutations compensating for the restoration of a gene (csrA), which is present in all or most of the different strains tested. This observation about the compensation after restoration of a a gene seem to not directly address the basic question of the paper (mutations beneficial after specific gene deletions).

The paper provides a detailed description of all mutations significantly associated with different ancestors and environments. At times, this extensive information feels like a long list of anecdotes, making it difficult to parse. I would prefer reducing this part to its essence, perhaps by focusing on the few most important specific mutations and summarizing the rest, relegating detailed information to the supplementary information (SI). Alternatively, mutations could be grouped into fewer classes, each described in detail.

Key to the paper are the association analyses, mostly relegated to the SI, which is common. However, given their importance, I would like to see a more detailed discussion of the statistical power to detect meaningful associations and how this informed the experiment's design. For instance, how does the power to detect associations depend on the number of replicates per ancestor? Was this used to decide on six replicates? Why was only one clone sampled from the endpoint? (I would have expected at least two endpoints to ensure variation is captured both between endpoints and within replicates.) What factors influence the statistical power of the analysis? Can you rule out that some associations of non-preexisting mutations arose from sharing a founding culture (e.g., via Luria-Delbrück effects, which can generate jackpot events)?

Other comments:

* It would be helpful to explain the rationale for using Acinetobacter baylyi. Is it particularly significant or convenient for experiments?

Figure 4:

* If strains are missing a horizontal line, does it mean they lack the corresponding gene?

* Does this graph show all detected mutations? If not, which ones are shown?

**Have all data underlying the figures and results presented in the manuscript been provided?**

Reviewer #1: Yes

Reviewer #2: Yes

Reviewer #3: Yes

PLOS authors have the option to publish the peer review history of their article (what does this mean?). If published, this will include your full peer review and any attached files.

Reviewer #1: No

Reviewer #2: No

Reviewer #3: No

---

## [Editor Report · Decision Letter 1]

5 Sep 2024

Dear Dr Barrick,

We are pleased to inform you that your manuscript entitled "Evolution recovers the fitness of *Acinetobacter baylyi* strains with large deletions through mutations in deletion-specific targets and global post-transcriptional regulators" has been editorially accepted for publication in PLOS Genetics. Congratulations!

Yours sincerely,

Xavier Didelot

Academic Editor

PLOS Genetics

Lotte Søgaard-Andersen

Section Editor

PLOS Genetics

Comments from the reviewers (if applicable):

**Data Deposition**

http://datadryad.org/submit?journalID=pgenetics&manu=PGENETICS-D-24-00557R1

**Press Queries**

---

## [Editor Report · Acceptance letter]

10 Sep 2024

PGENETICS-D-24-00557R1 

Evolution recovers the fitness of *Acinetobacter baylyi* strains with large deletions through mutations in deletion-specific targets and global post-transcriptional regulators 

Dear Dr Barrick, 

We are pleased to inform you that your manuscript entitled "Evolution recovers the fitness of *Acinetobacter baylyi* strains with large deletions through mutations in deletion-specific targets and global post-transcriptional regulators" has been formally accepted for publication in PLOS Genetics! Your manuscript is now with our production department and you will be notified of the publication date in due course.

With kind regards,

Zsofia Freund

PLOS Genetics

On behalf of:
